# Efficient Simultaneous Introduction of Premature Stop Codons in Three Tumor Suppressor Genes in PFFs via a Cytosine Base Editor

**DOI:** 10.3390/genes13050835

**Published:** 2022-05-07

**Authors:** Haoyun Jiang, Qiqi Jing, Qiang Yang, Chuanmin Qiao, Yaya Liao, Weiwei Liu, Yuyun Xing

**Affiliations:** State Key Laboratory of Pig Genetic Improvement and Production Technology, Jiangxi Agricultural University, Nanchang 330045, China; jianghaoyun1995@163.com (H.J.); jngqq2017@outlook.com (Q.J.); yangqiangjxau@outlook.com (Q.Y.); qiaochuanmin@hnaas.org.cn (C.Q.); avonliaoyaya@hotmail.com (Y.L.); weiweiliu1993@outlook.com (W.L.)

**Keywords:** base editing, triple-gene editing, pig, PFFs, hA3A-BE3-Y130F, mRNA-Seq

## Abstract

Base editing is an efficient and precise gene-editing technique, by which a single base can be changed without introducing double-strand breaks, and it is currently widely used in studies of various species. In this study, we used hA3A-BE3-Y130F to simultaneously introduce premature stop codons (TAG, TGA, and TAA) into three tumor suppressor genes, *TP53*, *PTEN*, and *APC*, in large white porcine fetal fibroblasts (PFFs). Among the isolated 290 single-cell colonies, 232 (80%) had premature stop codons in all the three genes. C−to−T conversion was found in 98.6%, 92.8%, and 87.2% of these cell colonies for *TP53*, *PTEN*, and *APC*, respectively. High frequencies of bystander C−to−T edits were observed within the editing window (positions 3–8), and there were nine (3.01%) clones with the designed simultaneous three-gene C−to−T conversion without bystander conversion. C−to−T conversion outside the editing window was found in 9.0%, 14.1%, and 26.2% of the 290 cell colonies for *TP53*, *PTEN*, and *APC*, respectively. Low-frequency C−to−G or C−to−A transversion occurred in *APC*. The mRNA levels of the three genes showed significant declines in triple-gene-mutant (Tri-Mut) cells as expected. No PTEN and a significantly lower (*p* < 0.05) APC protein expression were detected in Tri-Mut cells. Interestingly, the premature stop codon introduced into the *TP53* gene did not eliminate the expression of its full-length protein in the Tri-Mut cells, suggesting that stop codon read-through occurred. Tri-Mut cells showed a significantly higher (*p* < 0.05) proliferation rate than WT cells. Furthermore, we identified 1418 differentially expressed genes (DEGs) between the Tri-Mut and WT groups, which were mainly involved in functions such as tumor progression, cell cycle, and DNA repair. This study indicates that hA3A-BE3-Y130F can be a powerful tool to create diverse knockout cell models without double-strand breaks (DSBs), with further possibilities to produce porcine models with various purposes.

## 1. Introduction

The clustered regularly interspaced short palindromic repeat (CRISPR)/Cas9 technique is considered the most effective and convenient genome editing system to date. The guide RNA (gRNA) guides the Cas9 protein to produce a DNA double-strand break (DSB) at the target site, and the break is repaired through non-homologous end joining (NHEJ) or homology-dependent repair (HDR) pathways [1,2]. NHEJ pathways often induce insertion/deletion mutations (indels), causing the loss of protein function, although, sometimes, they can produce new functional proteins [3]. CRISPR/Cas9-mediated HDR can introduce point mutations or insertions using DNA donor templates [4]. However, HDR efficiency is usually much lower than that of NHEJ in mammalian cells [5], as an additional donor template is required for HDR repair. In addition, DSBs created by the Cas9 nuclease have frequently been found to result in genome instability, leading to undesired large deletions, translocations, and complex rearrangements [6,7,8,9]. In recent years, a modified CRISPR/Cas-based genome-editing tool, the cytosine base editor (CBE), has gained popularity [10,11]. The essence of the new system is to convert C base pairs in the editing window to T at the target site without introducing DNA DSBs [10,12]. The potential of the new tool has been demonstrated in endeavors to correct and introduce single point mutations, control gene splicing, and introduce in-frame stop codons [13,14,15,16].

The initial version of CBE, BE1, was produced by fusing rat APOBEC1 (rA1) with catalytically dead Cas9 (dCas9), with relatively low C−to−T editing efficiencies (0.8–7.7%) in mammalian cells [10]. Efforts have been made ever since to improve the efficiency and accuracy of base editing, leading to the emergence of many optimized versions of CBEs, such as BE3, BE4-Gam, AnCBE4max, hA3A-BE3, and hA3A-BE3-Y130F [17,18,19,20,21]. The hA3A-BE3-Y130F editor, which replaces the rat APOBEC1 in BE3 with human cytidine deaminase (Human_APOBEC3A_Y130F), was found to exhibit impressively high editing efficiencies in HEK293T cells [20] and the G/C−rich regions of the mouse genome of these optimized CBE versions [21].

## 2. Materials and Methods

### 2.1. Construction of Expression Vectors 

pCMV-hA3A-BE3-Y130F used in this study was reported previously [20] and purchased from Addgene (www.addgene.org/113428, accessed date 20 August 2018). pGL3-U6-sgRNA-PGK-puromycin used in this study [22] was reported previously and purchased from Addgene (www.addgene.org/51133, accessed date 2 March 2014). SgRNA oligonucleotides targeting the *TP53*, *PTEN*, and *APC* genes (Figure 1a–c) were designed using online software (http://crispor.tefor.net/, accessed date 20 August 2018), and annealed and cloned into the pGL3-U6-sgRNA-PGK-puromycin plasmid after BsaI (NEB, R0535S) digestion. All plasmid DNA was extracted using EndoFree^®^ Plasmid Maxi Kit (Qiagen, Hilden, Germany). 

### 2.2. PFF Culture and Transfection

PFFs from Large White fetuses on day 30 of gestation were cultured in Dulbecco’s Modified Eagle’s Medium (Gibco) supplemented with 10% fetal bovine serum (Gibco) and 1% penicillin/streptomycin (Gibco). Approximately 3 × 10^6^ PFFs were electroporated with 30 μg of the pCMV-hA3A-BE3-Y130F vector together with 10 μg each of the three sgRNA expression plasmids in 200 μL of Entranster-E (Engreen) using BTX ECM 2001 (Harvard Bioscience, Holliston, MA, USA). Approximately 3 × 10^6^ PFFs were transfected with 30 μg of the pCMV-hA3A-BE3-Y130F vector and 10 μg empty sgRNA expression plasmid to generate the WT group. The electroporation parameters for 2 mm-gap cuvettes were as follows: 210 V, 1 ms, and 3 pulses for 1 repeat. Cells were cultured for 6 h, replenished with new medium and cultured for another 30 h, and then the medium was replaced by fresh medium containing 2.5 μg/mL puromycin. After 2 days puromycin selection, the cells were plated into 10 cm petri dishes with various cell densities. After 6–8 days of culture, individual cell colonies were collected and cultured in 24-well plates. When the confluence reached about 95%, the cell colonies were sub-cultured in 6-well plates, and 25% of each cell colony was collected and lysed in 10 μL of lysis buffer (0.45% NP-40 plus 0.6% proteinase K) for 90 min at 56 °C, and then for 10 min at 95 °C. The lysate was then used as a template for PCR and subsequent sequencing. Eight colonies with genotypes *TP53*^−/−^, *PTEN*^−/−^, and *APC*^+/−^ (termed Tri-Mut colonies) were used for further analyses. Among these 8 colonies, 4 colonies were from 9 colonies that had desired stop codons in all three genes and without bystanders (Table 1), 3 had bystander conversion in *TP53*, and 1 had bystander conversion in *PTEN* (Appendix A). As our goal was to generate triple-knockout cells, these bystander conversions were deemed inconsequential.

### 2.3. Off-Target Analysis

The potential OTS for each sgRNA was predicted via an online design tool (http://crispor.tefor.net/, accessed date 2 July 2018) [23,24]. The top 10 potential OTSs for each gene of the selected 8 Tri-Mut single-cell clones were amplified using PCR and then subjected to Sanger sequencing. The sequences of each top 10 potential OTS for the three sgRNAs and primers used to amplify the OTSs are listed in Appendix A.

### 2.4. Cell Counting

For cumulative population doubling analysis, all 1.5 × 10^4^ Tri-Mut and WT cells from P3 to P21 were cultured in 6-well plates, passaged, and counted every 4 days. The population doubling time for each passage was calculated from the data of cell counts. To compare the relative proliferation rates, all 1.5 × 10^4^ Tri-Mut and WT cells from passage 4 were cultured in 6-well plates for 2, 4, and 6 days. Cells were harvested via trypsinization at various time points, stained with trypan blue, and counted on a hematocytometer under microscope.

### 2.5. Cell Proliferation Detected by CCK-8 kit

CCK-8 assay was implemented to assess the proliferation rate of Tri-Mut and WT cells. Cells of P17 Tri-Mut, P26 Tri-Mut, and P17 WT were seeded in five 96-well plates at a density of 5000 cells/well in 100 μL of culture medium, and culture medium without cells was used as a blank control. After 6, 24, 48, 72, and 96 h of culture, culture medium was replaced by 100 μL fresh medium plus 10% CCK-8 solution. The absorbance at 450 nm was read using an automatic microplate reader (TECAN, infinite M200 PRO, Switzerland). Each reaction was carried out in technical quadruplicates.

### 2.6. Tumorigenesis Assay In Vivo

To assess the tumorigenic potential of Tri-Mut cells, 2 × 10^6^ P17 Tri-Mut cells suspended in 200 μL phosphate-buffered saline (PBS) were transplanted subcutaneously into 5-week-old BALB/c nu/nu mice, and 2 × 10^6^ A375 cells in 200 μL PBS, 2 × 10^6^ A375 cells, and 1 × 10^6^ P17 Tri-Mut cells in 300 μL PBS were used as positive controls. Tumor volume of these cell-injected mice was measured weekly. The size of each tumor was calculated every week using the following formula: tumor volume(mm^3^) = longest diameter of tumor (mm) × shortest diameter of tumor (mm)^2^/2. The BALB/c nu/nu mice were purchased from GempharmatechCo., Ltd., Changzhou, China), and A375 cells were kindly provided by Stem Cell Bank, Chinese Academy of Sciences. Animals were cared for in compliance with the ARRIVE (Animal Research: Reporting of In Vivo Experiments) guidelines. 

### 2.7. RNA Extraction, Library Generation, and Sequencing

Total RNA from WT and Tri-Mut cells were collected using Trizol reagent (Invitrogen, Carlsbad, CA, USA) according to the manufacturer’s instructions, and RNA-seq was performed by Novogene BioTech Co., Beijing, China. Briefly, mRNA was first purified from the qualified RNA using poly-T oligo-attached magnetic beads. Fragmentation was carried out using divalent cations under elevated temperature in First Strand Synthesis Reaction Buffer (5×). Random hexamer primers and M-MuLV reverse transcriptase (RNase H) were used to synthesize first-strand cDNA, and second-strand cDNA synthesis was subsequently performed using RNase H and DNA polymerase I. After adenylation of 3’ ends of DNA fragments, adaptors with hairpin-loop structure were ligated to prepare for hybridization. The processed cDNA was PCR-amplified to construct cDNA libraries. Then, the library fragments were purified using the AMPure XP system (Beckman Coulter, Beverly, CA, USA) in order to select cDNA fragments of, preferentially, 370~420 bp in length. At last, library quality and integrity were assessed using the Agilent Bioanalyzer 2100 system, and the Illumina Novaseq platform was used to sequence the prepared cDNA libraries. 

### 2.8. Quality Control, Reads Mapping, and Quantification of Gene Expression Levels

Raw data (raw reads) in fastq format were firstly processed. In this step, clean data were obtained by removing reads containing adapters and greater than 10% of poly (N) and low-quality reads (>50% of the bases had Phred quality scores < 10). At the same time, the contents of Q20, Q30, and GC in the clean data were calculated. All the downstream analyses were based on the high-quality clean data.

### 2.9. Analysis of Differentially Expressed Genes

Differential expression analysis between Tri-Mut and WT cells was performed using DESeq2 software (version: 1.20.0) (https://bioconductor.org/packages/DESeq2/, Heidelberg, Germany, accessed date 20 August 2018) and a model based on a negative binomial distribution. Level and abundance of mRNA expression were normalized using the fragments per kilobase of transcript sequence per million base pairs sequenced (FPKM) approach based on the sequencing depth and gene length [25]. Differential expression analysis between the Tri-Mut and WT groups was implemented using the DESeq2 R package (version: 1.20.0) [26]. Genes with Benjamini–Hochberg-adjusted *p*-values < 0.05 and with |log2(fold change)| > 1.5 were defined as differentially expressed [27].

### 2.10. KEGG Enrichment Analysis of DEGs

Cluster Profiler R package [28] was used to analyze the statistical enrichment of DEGs in KEGG pathways, and corrected *p*-value < 0.05 was used as a threshold of significance for the enriched KEGG terms for the DEGs.

### 2.11. Validation of Gene Expressions Using RT-PCR Analysis and Western Blotting

Reverse transcription-polymerase chain reaction (RT-PCR) was conducted with primers listed in Appendix A. Fluoresce was monitored with the SYBR Green detection kit (Thermo Fisher Scientific, Waltham, MA, USA) using a 7900 Fast Real-Time PCR System (Applied Biosystems, Foster City, CA, USA). The thermal cycles were as follows: 94 °C for 10 min; 40 cycles of 95 °C for 15 s, and 60 °C for 1 min. Relative gene expression was calculated using the 2^−ΔΔCt^ method [29]. For Western blotting, total protein was extracted from the Tri-Mut and WT single clones with RIPA lysis buffer (Beyotime, Shanghai, China) containing 1 mM phenylonethysulfonyl fluoride (PMSF, Beyotime, Shanghai, China) at 4 °C. Protein concentration was determined using the bicinchoninic acid (BCA) assay, and 10 μg of protein from each sample was resolved by 10% sodium dodecyl sulfate-polyacrylamide gel electrophoresis (SDS-PAGE) (Angle Gene, Nanjing, China) and transferred onto polyvinylidene fluoride (PVDF) membranes (Millipore, Burlington, MA, USA). After blocking with Quick Block™ Blocking Buffer (Beyotime, Shanghai, China) for 1 h at room temperature, the membranes were washed in Tris-Buffered Saline with 0.1% Tween 20 (TBST) and incubated overnight at 4 °C with primary antibodies, including rabbit polyclonal antibodies against TP53 (1:500; ab131442, Abcam, Cambridge, UK) and β-actin (1:1000; ab8227, Abcam, UK), and rabbit monoclonal antibodies against PTEN (1:1000; ab170941, Abcam, UK) and APC (1:5000; ab40778, Abcam, UK). The membranes were washed and incubated with horseradish peroxidase (HRP)-conjugated secondary anti-rabbit antibody (1:10,000; ab6721, Abcam, UK) for 1h at room temperature. β-actin was used as the loading control. The bands were visualized with ECL in the ChemiDoc^TM^ MP imaging system (Bio-Rad, Hercules, CA, USA). Band intensities were quantified using ImageJ (Rawak Software, Germany). The relative amounts of TP53, PTEN, and APC were calculated after normalization to β-actin.

### 2.12. Protein–Protein Interaction (PPI) Network Analysis

The top 100 DEGs were analyzed to unfold their interactions using the STRING version 11.5 database (available at: https://cn.string-db.org, accessed date 12 August 2021) [30]. K-means clustering, which is an unsupervised clustering algorithm based on the adjacency matrix, was used to group molecules based on pre-specified criteria. Cluster edges are colored differently to represent the different sources of the interaction data.

### 2.13. Statistical Analysis

Data are presented as means ± SEM. Statistical significance was assessed using the independent sample Student’s *t*-test and the SPSS v20.0 software package (IBM SPSS, Chicago, IL, USA). Data are considered statistically significant when *p*-value < 0.01 indicates that the difference is extremely significant.

## 3. Results

### 3.1. Efficient Base Editing of Multiple Genes in PFFs

In our study, premature stop codons were generated by a single C−to−T conversion in each of the three tumor suppressor genes in PFF cells via three sgRNA expression vectors targeting these three genes and hA3A-BE3-Y130F (Figure 1). After single-cell isolation, 290 single-cell clones were obtained. PCR and Sanger sequencing were employed to detect the C−to−T conversion in these colonies. The sequencing results showed that 286 had the C > T conversion at the target site for *TP53*, 269 for *PTEN*, and 253 for *APC*. The majority of *TP53* and *PTEN* mutant clones were homozygous, while the majority of *APC* mutants were heterozygous (Table 1). Among these 290 colonies, premature stop codons were simultaneously introduced into 232 colonies (80%) for all the three genes (Table 1). 

However, unintended bystander conversions, in and/or outside the window of C−to−T substitutions other than the target C−to−T, also occurred at significant rates. Unintended C > T conversion was collectively found in 186 colonies (64.14%) at positions 1, 3, 11, and/or 13 for *TP53*; in 218 colonies (75.17%) at positions −1, 1, 2, 7, and/or 8 for *PTEN*; and 78 colonies (26.89%) at position 9 for *APC* (Figure 2b–e). It should be mentioned that only nine single clones had precisely the designed stop codons in all the three genes with no bystander editing (Table 1). Moreover, low-frequency unintended C−to−G, C−to−A, and G−to−C base transversions were found in *APC* (Figure 3c). Indels were detected in three single clones at the *PTEN* locus (1.03%) and in two clones at the *APC* locus (0.69%) (Table 1, Figure 3b,c). For further study, we chose eight colonies that had the desired homozygous stop codons in *TP53* and *PTEN*, and the heterozygous stop codon in *APC*. We term these colonies (*TP53*^−/−^, *PTEN*^−/−^, and *APC*^+/−^) as triple-gene-mutant (Tri-Mut) cells. The top ten predicted off-target sites (OTSs) for each of the three genes were examined for these eight Tri-Mut colonies using PCR and Sanger sequencing. No off-target event was found at these predicted OTSs (Appendix A). Furthermore, RT-PCR was performed to determine the mRNA levels of *TP53*, *PTEN*, and *APC* in these Tri-Mut colonies. Compared with WT cells, the mRNA levels of *TP53*, *PTEN*, and *APC* showed 94%, 84%, and 64% decreases (*p* < 0.0001), respectively, in these Tri-Mut cells (Figure 4a–c). We performed Western blot analysis of the Tri-Mut single clones to confirm the effect of the disruption of these three genes. The results showed that no PTEN protein was detected and that the expression of the APC protein was significantly low (*p* < 0.05) (Figure 4e,g,h). Interestingly, we observed the expression of full-length TP53 in Tri-Mut single clones, which contain the nonsense mutant (Figure 4d–h), and more interestingly, the protein level was higher than WT even though the mRNA level was significantly lower (Figure 4a,d,f).

### 3.2. Proliferation and Tumorigenicity Testing of Tri-Mut Cells

The doubling time assay indicated that the proliferation rate of Tri-Mut cells was significantly higher (*p* < 0.001) than that of WT cells (Figure 4i). The growth rates of passage 4 (P4) Tri-Mut cells were significantly (*p* < 0.001) higher than those of P4 WT cells (Figure 4j). The cell counting kit-8 (CCK-8) assay revealed that the proliferation rates of randomly selected P17 and P26 Tri-Mut cells were significantly higher (*p* < 0.01) than those of WT cells (P17; Figure 4k). However, Tri-Mut cells showed no tumorigenic potential, nor did they enhance the tumorigenicity of A375 cells (Figure 5).

### 3.3. Identification of DEGs in Tri-Mut Cells

In the present study, eight Tri-Mut single-cell clones and seven WT cell colonies were used to generate mRNA sequence (mRNA-Seq) data. There was an average of 41570875 and 42202074 clean reads (150 bp, pair-end) with an average of 6.5 G and 6.6 G clean bases generated using the Illumina NovaSeq platform, of which 96.4% and 96.6% were mapped to the Sus scrofa 11.1 reference genome for Tri-Mut cells and WT cells, respectively. Principal component analysis based on fragments per kilobase of transcript sequence per million base pairs sequenced (FPKM) values were performed to assess the mRNA-Seq data, and the results suggest that these two groups underwent strong separation (Appendix A). 

We further conducted a differential expression analysis on the two groups. As shown in Figure 6a, 595 and 403 genes were exclusively expressed in Tri-Mut and WT cells, respectively. A volcano plot (Figure 6b) was used to infer the overall distribution of the DEGs. There was a total of 1418 DEGs between the Tri-Mut and WT groups, in which 894 genes were up-regulated and 524 genes were down-regulated in Tri-Mut. The top 30 DEGs are listed in Appendix A. As shown in this table, most genes are significantly associated with tumor progression or the cell cycle pathway 

### 3.4. Functional Enrichment Analysis of DEGs of Tri-Mut Cells

Signaling pathway enrichment analysis based on KEGG was performed to analyze the functional impacts of the DEGs. Twenty-two significantly enriched KEGG pathways were identified *(p*-adjust < 0.05), of which most enriched pathways are significantly associated with tumor progression and/or cell proliferation and differentiation (Figure 6c).

### 3.5. Protein–Protein Interaction (PPI) Networks of Top 100 DEGs between Tri-Mut and WT Groups

We used STRING to perform PPI analysis of the top 100 DEGs between Tri-Mut and WT groups (Appendix A). K-means clustering was used to obtain a better representation of the protein interactions, and the network was divided into three clusters. The total number of nodes was 100, the total number of interactions was 132, and the PPI enrichment *p*-value was 1.0 × 10^−16^. In the network, TP53, PTEN, CDKN1A, MDM2, MDM4, BBC3, POLK, RPS6KA2, BCL2L1, TGFBR2, CCND1, and GDF15 were present as pivotal nodes, of which TP53 was highly interconnected with other nodes. 

### 3.6. Validation of DEGs via RT-PCR

We randomly selected 13 up-regulated (*PTGFRN*, *COL9A3*, *SCD*, *THYN1*, *DCLK1*, *SH2D5*, *MFSD2A*, *CRABP2*, *ALX1*, *S100A10*, *CMKLR1*, *VASH2*, and *SOX9*) and 6 down-regulated genes (*KIF12*, *ZMAT3*, *XPC*, *EIF1AY*, *CCDC50*, and *INKA2*) for the RT-PCR to validate the DEGs from the mRNA-seq data. The relative expression of these genes detected using RT-PCR had the same tendency of changes as found in the RNA-seq analysis (Appendix A), indicating that the sequencing data were accurate and reliable.

## 4. Discussion

As a powerful genome editing tool, the CRISPR/Cas9 system induces DSBs at target sites specified by sgRNAs, and indel mutations are induced via the dominant NHEJ pathway [2]. However, there have been concerns about Cas9-mediated DSBs, including the production of novel functional proteins and genome destabilization [3,4,5]. In recent years, two innovative base editing tools based on the CRISPR/Cas9 system, namely, CBEs and adenine base editors (ABEs), have been developed to directly change single nucleotides without requiring DSBs or donor DNA [11]. CBEs induce C−to−T conversions in the activity window and, thus, can produce premature stop codons by converting a CAA/CAG/CGA codon into TAA/TAG/TGA within a coding exon. In our previous study, we created *P53* KO and point-edited PFFs with considerable efficiencies via CRISPR/Cas9 technologies [31]. In the present study, we chose to simultaneously introduce premature stop codons in three well-known tumor suppressor genes, *TP53*, *PTEN*, and *APC* [32,33,34], in PFFs, using the hA3A-BE3-Y130F system, which had previously shown impressively high editing efficiency in human cells [20].

We totally isolated 290 single-cell colonies after the co-transfection of the hA3A-BE3-Y130F vector and three sgRNA vectors targeting *TP53*, *PTEN*, and *APC*. PCR and sanger sequencing showed that 98.6%, 92.8%, and 87.2% of the single-cell colonies harbored desired stop codons in *TP53*, *PTEN*, and *APC* genes, respectively. In addition, 80.0% of the single clones simultaneously contained desired stop codons in all these three genes (Table 1). In a recent report [35], hA3A-BE3-Y130F exhibited around 10–60% C−to−T editing efficiencies in PFFs for a three-gene simultaneous-editing experiment, and showed 55.6–100% editing efficiencies for single and multiple genes in porcine embryos. In our study, this CBE displayed 87.2–98.6% desired C−to−T mutation rates in each of the three tumor-suppressor genes. We believe that the appreciably higher editing efficiencies in our study were primarily due to the 48h puromycin selection. Another two recent studies showed 88.98% median single-gene editing frequencies in mouse embryos [21] and displayed 10–60% C−to−T mutation rates at GpC sites in methylated regions of different single genes in 293T cells using the hA3A-BE3-Y130F editor [20]. In comparison with other CBEs that have been used in PFFs, hA3A-BE3-Y130F also exhibited higher editing efficiency in our study. BE3 showed 12–84% single-gene base editing rates in another report [36], and it displayed 0–21.6% single-gene mutation rates and 1.5–25.2% multiple-gene mutation rates in another study [37]. hA3A-BE3 has been demonstrated to result in 40% and 55% rates in two separate three-gene simultaneous-editing experiments [37]. hA3A-BE3-NG and BE4Gam exhibited 25.93% and 7.5% three-gene mutation rates in PFFs in two studies, respectively [38,39]. Our study indicates that hA3A-BE3-Y130F has excellent C−to−T conversion efficiency for single and multiple genes in PFFs. 

It has been shown that bystander editing by CBEs in the editing window is common [40]. In recently reported studies, hA3A-BE3-Y130F generated around 10% bystander editing in U2OS cells [41] and up to 83.35% bystander editing in different genes in porcine embryos [35]. In the present study, a high frequency of unintended C−to−T editing occurred in all the three genes (Table 1). Moreover, around 7% and 60% single-cell clones contained C−to−T substitutions for more than two positions in *TP53* and *PTEN* genes (Figure 3). It should be noted that, although high-frequency bystander C−to−T edits were identified, there were still nine (3.01%) single-cell clones that only had the designed simultaneous C−to−T conversions (Table 1), suggesting that hA3A-BE3-Y130F can be a powerful tool for defined C−to−T editing. However, in studies similar to the present one, most bystander mutations are inconsequential, as premature stop codons are introduced [40]. 

A typical CBE editing window has an average C−to−T conversion rate exceeding 40% (0.4) [42,43]. The editing window of hA3A-BE3-Y130F was initially defined as spanning positions 3–8 (C3–C8) within the protospacer, and 2–40% bystander editing efficiencies for C2 and C9–C17 were observed in different gene regions [20]. In our study, the average C3–C8 conversion efficiencies exceeded 80% (Figure 2). In addition, hA3A-BE3-Y130F exhibited sequence preferences in the order of TC > CC ≥ AC > GC, consistent with the first version of CBEs [10]. Unexpected byproducts, including indels and non-C−to−T conversions, have also been observed when using CBEs [40]. Regarding the hA3A-BE3-Y130F editor, varied non-C−to−T and indel frequencies were found for single-gene editing in 293T cells, and mouse and porcine embryos [20,21,35]. In this study, five single-cell colonies (1.72%) carried indels (three for *PTEN* and two for *APC*), and four single-cell colonies (1.37%) had non-C−to−T conversions in the *APC* gene. 

The results of the RT-PCR showed that the levels of the RNA expression of these three genes decreased compared with WT, which is consistent with the genotype of edited genes (Figure 4a–c). Western blot analysis of PTEN and APC proteins confirmed the effect of the disruption of these two genes (Figure 4e). Surprisingly, we found that TP53, containing the nonsense mutation (c.R206*), expressed its full-length protein in Tri-Mut single clones (Figure 4d); this suggests that the stop codon read-through for TP53 occurred in these cells. Premature stop codon (PTC) read-through has been reported in other studies [44,45], and various factors can influence the stop codon read-through efficiency [46]. It is not clear whether the expressed mutant TP53 has the same functions as WT, and it is also puzzling that the expression level of the mutant TP53 protein was higher than that in WT cells, even though the mRNA was reduced to 8.4% of the WT level (Figure 4a). Our study suggests that, when introducing PTCs, Western blotting should always be required to measure gene expression.

It is well known that tumor-suppressor genes (TSGs) are involved in cell cycle regulation, cell differentiation, and cancer development [47,48]. In our previous study, *P53*-KO and *P53* point-edited PFFs showed significantly higher (*p* < 0.01) proliferation rates than *P53*-WT cells, and global changes in mRNAs and miRNAs were identified [31]. As expected, the Tri-Mut cells in this study showed a significantly higher proliferation ability (*p* < 0.01) (Figure 4i–k) than WT cells. Increased proliferation capacities were also found in *TP53*-KO human embryonic stem cells [49], *TP53*-KO canine fetal fibroblasts [50], *PTEN*-KO human MCF7 cells [51], and *TP53* and *PTEN* double-null mouse embryonic fibroblasts (MEFs) [52]. 

Moreover, we assessed the tumorigenicity of the Tri-Mut cells via nude mice in vivo. The Tri-Mut PFFs in this study did not show direct tumorigenic potential, nor did they enhance the tumorigenic effect of A375 cells up to 4 weeks after subcutaneous injection (Figure 5). The development of cancer or a tumor is a complex, multistep process. In addition to gaining uncontrolled division and proliferation abilities, tumor or cancer cells survive a variety of metabolic stresses, and complex regulation pathways are involved [53]. In many studies where tumors were highly expected in animal models, no tumor phenotypes or tumorigenic signs were detected [54,55,56,57]. The type of animal used for transplant and their immunosuppressive level [58], the number of transplanted cells, the time of the tumor vivo experiment, the complex gene regulatory network, etc., may lead to the diminishing of the tumorigenic potential and cause the insensitivity of the vivo tumorigenicity assay. The unexpected stop codon read-through in the *TP53* gene (and the higher level protein expression) was likely another important factor leading to the lack of tumorigenicity of the Tri-Mut cells.

Nevertheless, RNA-seq identified 1418 DEGs between the Tri-Mut and WT cells (Figure 6b), and KEGG enrichment analysis revealed that the DEGs were significantly associated with tumor progression, and cell growth and differentiation (Figure 6c). This indicates that different regulatory mechanisms of these three genes may have been involved in the PFFs, and the tumorigenic potential of the Tri-Mut cells obtained in this study is worthy of further investigation.

## 5. Conclusions

In conclusion, we successfully obtained *TP53*/*PTEN*/*APC* Tri-Mut PFFs with a remarkable high efficiency by introducing premature stop codons using hA3A-BE3-Y130F. The Tri-Mut cells exhibited the anticipated changes in gene expressions, cell proliferation capabilities, and alteration of regulatory pathways. Our study suggests that the hA3A-BE3-Y130F editor could be a powerful and reliable tool for C−to−T editing in PFFs.

## Figures and Tables

**Figure 1 genes-13-00835-f001:**
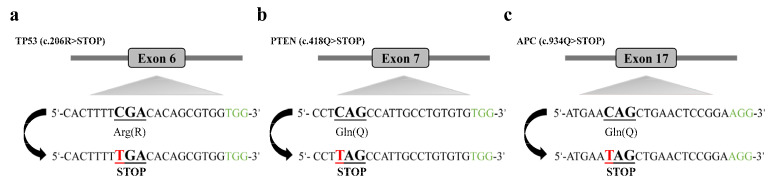
The target sequences at the *TP53* (**a**), *PTEN* (**b**), and *APC* (**c**) loci. PAM sequences are added to the 3′ end (marked in green); codons with intended C−to−T conversion are underlined; and the C to be converted (to T) is indicated in red.

**Figure 2 genes-13-00835-f002:**
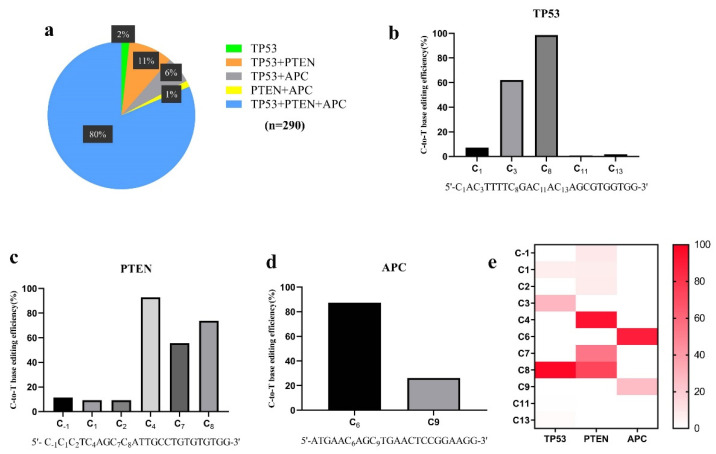
hA3A-BE3-Y130F-medidated C−to−T conversion in three tumor suppressor genes in PFFs. (**a**) Distribution of colonies harboring various combinations of mutations among the obtained 290 colonies. (**b**–**d**) Histograms of C−to−T base-editing efficiencies of three genes at different positions in the editing windows. (**e**) Heatmap of C−to−T base-editing efficiencies of three genes at different positions in the editing windows.

**Figure 3 genes-13-00835-f003:**
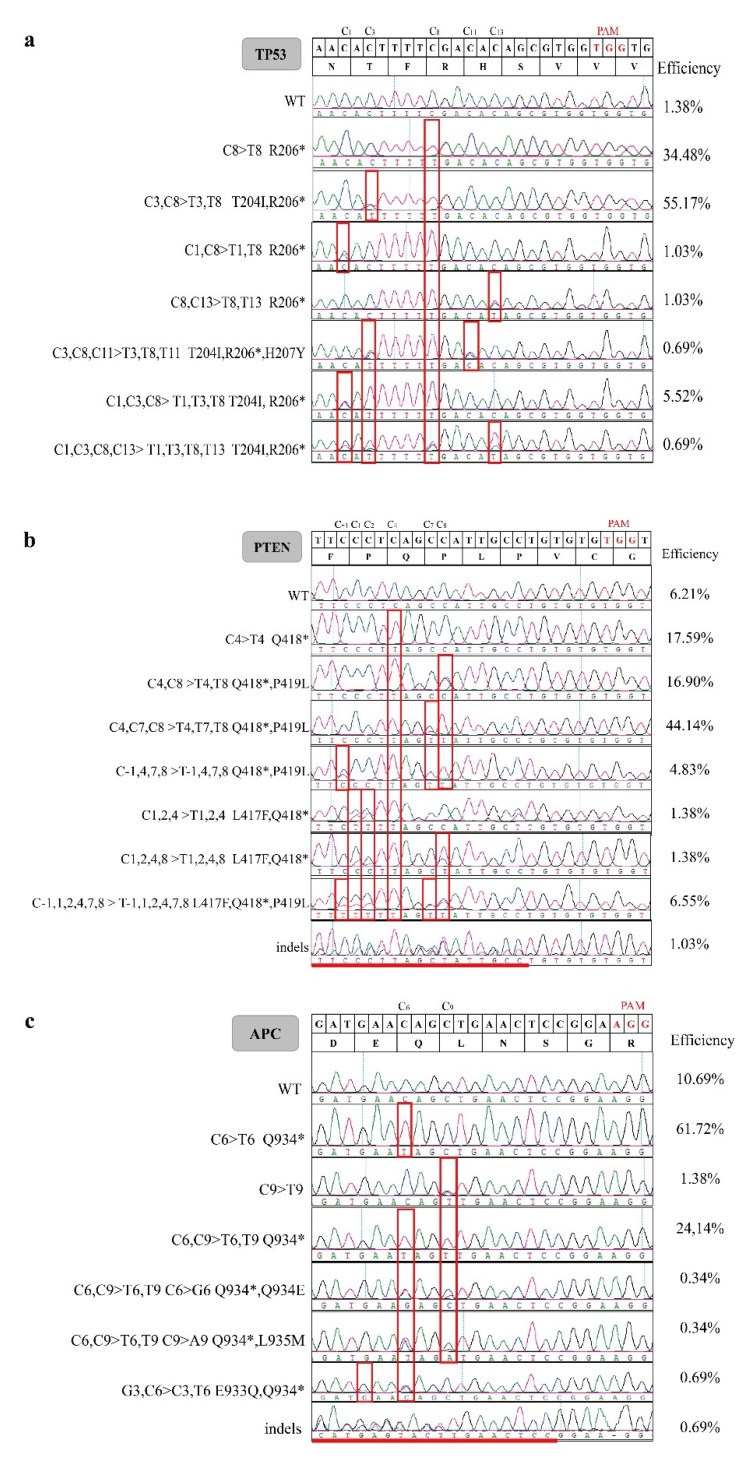
Base substitutions induced by the hA3A-BE3-Y130F system. (**a**–**c**) Sanger sequencing chromatograms of single-cell colonies showing gene modifications at *TP53* (**a**), *PTEN* (**b**), and *APC* (**c**) loci. The base substitutions are highlighted in red boxes and * represents stop codons.

**Figure 4 genes-13-00835-f004:**
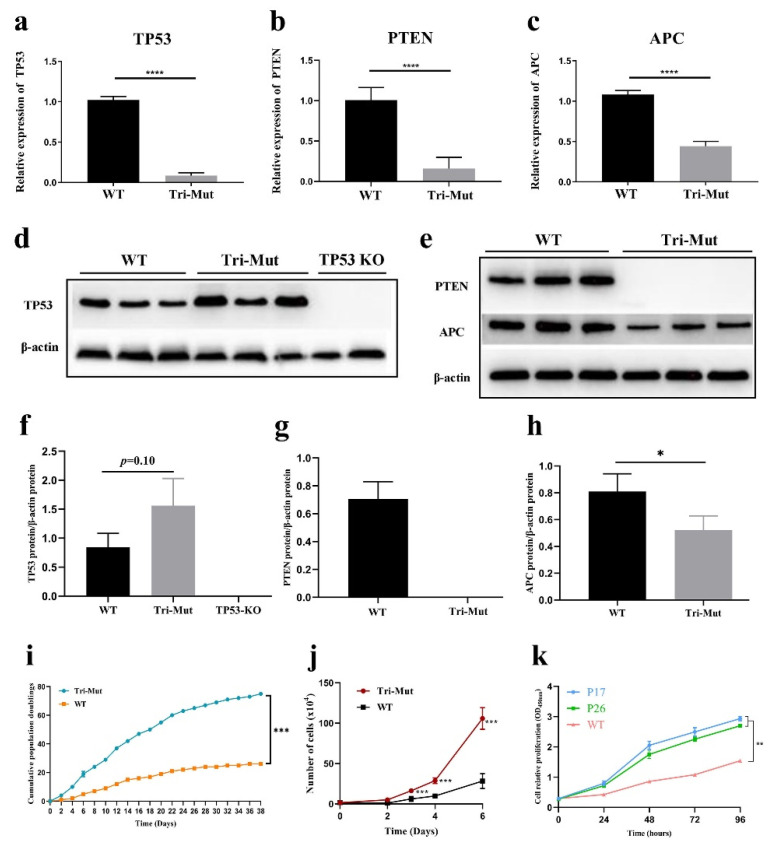
RT-PCR detection and Western blot analysis of the three targeted genes and proliferation ability analyses of Tri-Mut cells. The relative expressions of *TP53* (**a**), *PTEN* (**b**), and *APC* (**c**) detected with the qPCR assay. Western blot analyses of TP53 (**d**), PTEN, and APC (**e**). The *TP53* KO cells were from our previous study [31]. Quantitative analyses of protein levels of TP53 (**f**), PTEN (**g**), and APC (**h**). (**i**) Comparison of the population doublings of Tri-Mut and WT cells. (**j**) Comparison of relative proliferation rates of P4 Tri-Mut cells and WT cells (×10,000 cells) per well in 6-well plates. (**k**) Proliferation profiles of P17 and P26 Tri-Mut cells and P17 WT cells detected with CCK8 kit. Statistical analyses highlight the significant difference between Tri-Mut and WT cells. Data were analyzed by Student’s *t*-test (* *p* < 0.05; ** *p* < 0.01; *** *p* < 0.001; **** *p* < 0.0001; ns, not significant) and shown as means ± SEM (*n* = 3 independent experiments).

**Figure 5 genes-13-00835-f005:**
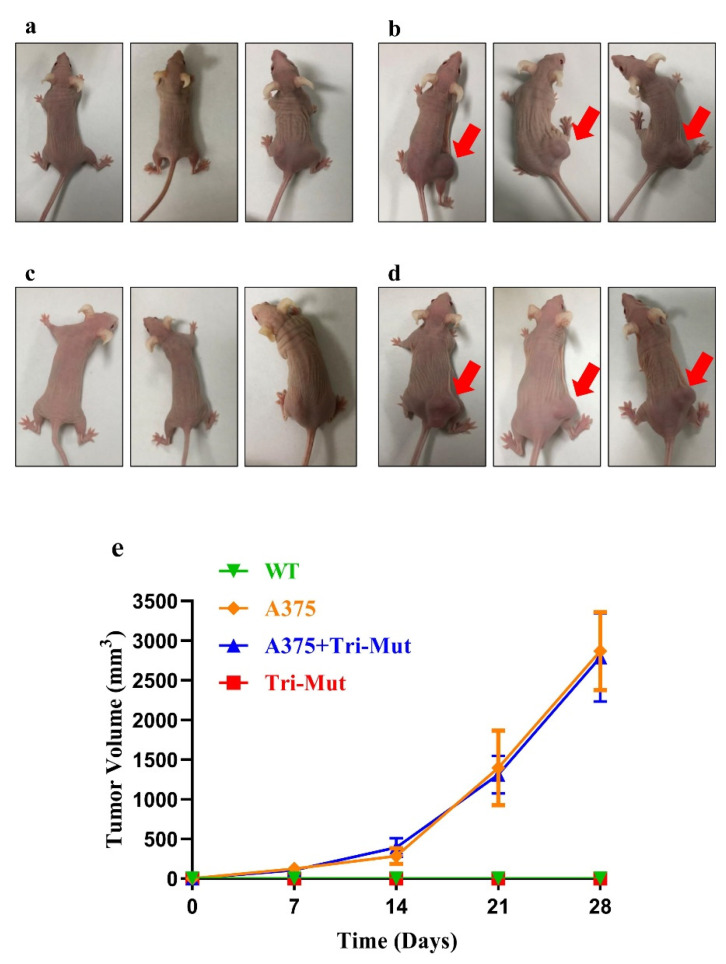
Subcutaneous tumorigenicity test with BALB/c nu/nu mice. Images of mice 28 days after subcutaneous injection: (**a**) control group (without injection), (**b**) A375-injected mice, (**c**) Tri-Mut cell-injected mice, (**d**) A375 + Tri-Mut cell-injected mice. (**e**) Tumor volume at 4 time points after subcutaneous injection in different groups.

**Figure 6 genes-13-00835-f006:**
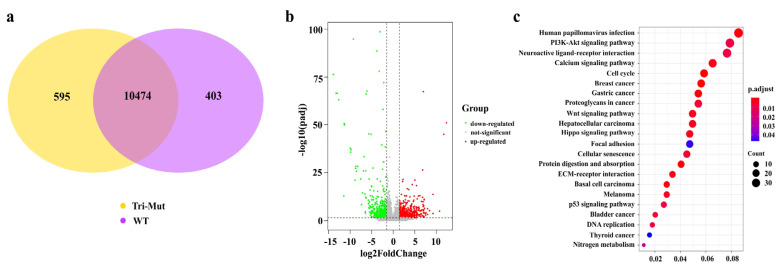
Venn diagram, volcano plot, and KEGG enrichment of DEGs between Tri-Mut and WT cells. (**a**) The Venn diagram shows the number of genes differentially expressed within groups. (**b**) Volcano map of differentially expressed genes between Tri-Mut cells and WT cells. The *x*-axis represents the value of log2 (fold change), and the *y*-axis shows the value of −1og10 (padj). (**c**) Statistics of KEGG pathway enrichment. The *x*-axis shows the enrichment factor; the *y*-axis lists KEGG pathways. The color of the dot denotes *p*.adjust, and the size of dots represents the number of DEGs mapped to the reference pathway. R package VennDiagram (1.6.19) was used to draw the Venn diagram of the number of DEGs in Tri-Mut and WT; R package ggplot2 (3.3.1) was used to draw the volcano map of analysis of DEGs and the bubble diagram of KEGG enrichment analysis.

**Table 1 genes-13-00835-t001:** Profile of the 290 PFF clones obtained from base editing using pCMV-hA3A-BE3-Y130F and 3 sgRNAs.

Target Genes	No. of Clones with Intended C−to−T Conversion (Homo/Het)	No. of Clones with Unintended C−to−T Conversion (%)	No. of Indels (%)	No. of Clones with Desired Stop Codons in All Three Genes (%)	No. of Clones with Desired Stop Codons in all Three Genes and without Bystanders (%)
*TP53*	286 (267/19)	186 (64.14)	0/290 (0)	232 (80.00)	9 (3.01)
*PTEN*	269 (262/7)	218 (75.17)	3/290 (1.03)		
*APC*	253 (52/201)	78 (26.89)	2/290 (0.7)		

## Data Availability

The RNA-seq raw datasets generated and analyzed during the current study are available in the figshare repository, https://figshare.com/s/4372d327371dec546c7e (accessed date 18 January 2022).

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
