# Peer review of "Efficient Simultaneous Introduction of Premature Stop Codons in Three Tumor Suppressor Genes in PFFs via a Cytosine Base Editor"

_genes, 2022, doi:10.3390/genes13050835_

Round 1
Reviewer 1 Report
Genome editing technologies, including base editing, have transformed functional genetics and genomics and are uniquely promising to understand cancer gene function, particularly in cell types and cell lines that have been recalcitrant to other types of genome engineering methods.
The authors in this study use porcine fibroblasts and introduce simple STOP codon mutations in three tumor suppressor genes (TSGs) — p53, Apc, and Pten — and they perform superficial characterization of the phenotypic consequences of these mutations. The results are straightforward and expected — mutation of these established TSGs promotes cellular proliferation and oncogenic transformation, as shown in vitro and in vivo. The results are not surprising, thus the novelty is low. The technology has been established for some time now, and so a high novelty study would be expected to delve deeper into either/or the biology and technology. This study falls short of that, but provides a nice technical overview of how these types of experiments are and can be performed and optimized.
As the results are straightforward, I have no major comments other than:
- From the RNA-seq data, do the authors see RNA editing in any of these three genes?
- The authors should delve deeper into the RNA-seq data and place it in the context of the cancer literature. What are the oncogenic transcriptional changes that these mutations are eliciting and how do they fit into the biology of each of these TSGs?
- Mutation of these TSGs transforms these cells. Is sustained inactivation required for the phenotype? The authors should re-express each of these genes in these cells and measure proliferation, at a minimum, to test this concept given the lack of overall novelty. I think this is a key experiment to do for this manuscript.
- STOP codons are only one type of alteration and, for some cancer genes, it is not the most common event. It would be interesting for the authors to test other types of mutations in their system and test whether they see different phenotypes depending on the type of mutation.
Reviewer 2 Report
The manuscript by Haoyun et al reports a high-efficiency method to generate point mutations in triple tumor-suppression genes simultaneously by hA3A-BE3-Y130F in PFFs, the resulting mutant cells show increased proliferation and distinct gene expression profiles compared to WT cells. Overall, this is a very straightforward study that shows the great potential of base editing in pig primary cells by hA3A-BE3-Y130F. However, a few major concerns need to be addressed.
- Did the wild-type cells used in this study undergo puromycin selection as the mutant cells do? It’s better to use the same cell lines that show negative editing for those 3 genes as wild-type cells for proliferation assay.
- Did the author test different doses of the hA3A-BE3-Y130F plasmid? A higher dose might induce higher bypass or other mutations.
- The author should discuss more why the mutant cells showed no tumorigenic potential.
Minor:
Line 36, induces should be induce.
Results 3.3 part is duplicated somehow.
Reviewer 3 Report
The present work is one more work regarding gene editing emphasizing CBE methodology to create premature stop codon in the frame of the gene of the protein. Here they have chosen three tumor suppressors genes TP53, PTEN, and APC. The work is well addressed, despite clearly, it will need further investigations to get an understanding of some physiological occurrences since all the characteristics of created cells (Tri-Mut PFF cells) had tumors profile, but they did not show it in vivo analysis. That is an interesting question brought up by this paper and would be nice to answer in the future. Regarding the methodology carrying modification in three different genes, even with a high number of bystanders, the authors have got a great success, getting 9 clones with pursued mutations and without bystanders. Besides they have shown different parameters that follow logical reasoning describing well their goals. Just would like to mention how they decided to show the functionality of the premature stop codon (Validation of gene expression). They have shown levels of specific RNA from mutated protein decreased compared with WT. Since the genes have mutated changing just one base pair in frame; how do the authors explain it affects the level of RNA expression/transcription of the gene? Would not it be mandatory to use protein-blot analysis? The work is meanly about the methodology, so this concern would be nice to be assessed in the discussion.
Round 2
Reviewer 1 Report
The authors have assessed some of my concerns, namely looking at RNA-seq data to look for RNA editing. The rest of the concerns were not addressed, but I think based on the main technical message that wants to be conveyed through this paper, I think it is fine to accept as is.